

# FoxO1 is a critical regulator of hepatocyte lipid deposition in chronic stress mice

Yun-zi Liu[1,*], Wei Peng[1,*], Ji-kuai Chen[2], Wen-jun Su[1], Wen-jie Yan[1], Yun-xia Wang[1] and Chun-lei Jiang[1]

[1] Department of Stress Medicine, Faculty of Psychology, Second Military Medical University, Shanghai, China
[2] Department of Health Toxicology, Faculty of Naval Medicine, Second Military Medical University, Shanghai, China
* These authors contributed equally to this work.

Corresponding author
Chun-lei Jiang, cljiang@vip.163.com

## ABSTRACT

Forkhead box O1 (FoxO1) is involved in lipid metabolisms. However, its role in chronic stress-related nonalcoholic fatty liver disease (NAFLD) is unclear. The scientific premise of our study was based on the finding that FoxO1 expression is increased in the liver of mice after chronic stress. It is important to understand the mechanisms involved in the activation of FoxO1 and how its function affects the liver lipid deposition. We employed a murine chronic stress model, in which mice were treated by plantar electrical stimulation and restraint for 6 weeks, and a cellular model, in which Hepa1–6 cells were treated with corticosterone. We also used a pharmacologic approach as1842856, a highly specific FoxO1 inhibitor. Lipid metabolism related genes levels were measured by qRT-PCR and the lipid levels by biochemical detection. We show that the level of FoxO1 is significantly elevated in the liver of chronic stress mice. Transcription factor FoxO1 regulates a lipid synthesis phenotype of hepatocyte that is involved in the development and progression of NAFLD. We have shown that inhibition of FoxO1 induced phenotypic conversion of hepatocytes and down-regulates lipid synthesis genes expression by hepatocytes, which contribute to lipid deposition in NAFLD. At the cellular level, the inhibitor of FoxO1 as1842856 can also attenuate the lipid deposition of Hepa1–6 cells induced by corticosterone. Targeting FoxO1 is a novel therapeutic target for chronic stress-related NAFLD.

## INTRODUCTION

The incidence of nonalcoholic fatty liver disease (NAFLD) is 20–30% globally (*Henao-Mejia et al., 2012*). The lipid deposition of the liver, especially the abnormal deposition of triglycerides (TG), is the basis of the pathogenesis of NAFLD (*Piccinin, Villani & Moschetta, 2018*). The correlation between the occurrence of NAFLD and chronic stress has received increasing attention in recent years (*Ashtari, Pourhoseingholi & Zali, 2015*). Our previous result and others' studies have confirmed that chronic stress could lead to liver lipid deposition and inflammation, which lead to the development of

NAFLD (*Fu et al., 2010*; *Liu et al., 2014*). However, the specific mechanism of chronic stress-induced liver lipid deposition is not clear.

Forkhead box O1 (FoxO1) is a member of the forkhead O family and plays a key role in lipid metabolism (*Li et al., 2017*). Activation of FoxO1 was detected during lipid deposition in the liver (*Kim et al., 2016*); activated FoxO1 was also found to lead to an increase in the output of very low-density lipoprotein (triglyceride-rich particles) in the liver, resulting in hypertriglyceridemia (*Kim et al., 2011*); the expression and activity of FoxO1 were increased in patients with NAFLD and correlated positively with the severity of the disease (*Valenti et al., 2008*). *Matsumoto et al. (2006)* found that the transfection of FoxO1 in the liver increased the TG content of the liver. Thus, we hypothesized that chronic stress leads to liver lipid deposition by activation of FoxO1. Our previous study demonstrated that chronic stress induces liver lipid deposition and FoxO1 activation in mice (*Liu et al., 2014*). However, the causal relationship between FoxO1 and liver lipid deposition during chronic stress is unclear. The molecular mechanisms of stress-induced lipid metabolism disorders are also unclear.

In this study, 6 weeks of plantar electrical stimulation and restraint were used to induce the mouse chronic stress. FoxO1 activity inhibitor as1842856 was used to verify the role of FoxO1 in this model. After 6-week stress exposure, we performed lipid deposition tests and measured TG synthesis genes and cholesterol metabolism-related genes to explore if and how FoxO1 influences the lipid deposition in a chronic stress animal model. It may provide new targets for drug prevention and treatment of stress-related lipid metabolic disorders.

# MATERIALS AND METHODS

## Reagents and antibodies

Forkhead box O1 inhibitor (as1842856) and Cortisol were purchased from Sigma-Aldrich (St. Louis, MO, USA). Fetal calf serum was obtained from Hyclone (Logan, UT, USA). Penicillin and streptomycin were obtained from Invitrogen (Carlsbad, CA, USA). FoxO1, *p*-FoxO1, and β-Actin antibodies were obtained from Abcam (Cambridge, MA, USA). IRDye 800CW donkey anti-goat and IRDye 800CW goat anti-rabbit secondary antibodies were both purchased from LI-COR Biosciences (LI-COR, Inc., Lincoln, NE, USA). The Real-time PCR reactions were performed using SYBR Premix Ex Taq™ (Takara Biotechnology, Tokyo, Japan). Hematoxylin–eosin (H&E) and oil red O were purchased from Nanjing Jiancheng Science and Technology Company, Nanjing, China.

## Animals and chronic stress protocols

C57BL/6J male mice, 5–7 weeks age, were purchased from Slaccas Laboratory Animal Company (Shanghai, China). Mice were housed in plastic home-cages in a temperature controlled room at 24 °C, under a 12:12 h illumination cycle (lights on at 8:00 AM). To stress animals, we used a modified version of the protocol described previously (*Liu et al., 2014*). For chronic stress, each mouse was administered electric foot shock (10:00, 7.5 s/2 min, 15 min, 25V) and restraint stress (19:00–21:00) every day for 42 consecutive days before evaluations. Non-stress mice (control group) remained in the

home cage and kept isolated from stressed animals. FoxO1 inhibitor group were administered as1842856 at 30 mg/kg by gavage during the 42 days. The control group was treated with equal volume solvent by gavage. Animal's weight and food intake were measured every week. All animal procedures used in this study were approved by the Institutional Animal Care and Use Committee of the Second Military Medical University (No. 20161027; Shanghai, China) and Shanghai Science and Technology Committee (SYXK-HU-2012-0003).

## Hematoxylin–eosin and oil red O staining

Following fixation of the livers or cells with 10% formalin, livers were sliced and stained with H&E for histological examination. Hepatic lipid content was also determined by staining with Oil Red O (Sigma, St. Louis, MO, USA). Pictures were imaged with a Zeiss microscope (Carl Zeiss Microscopy, Thornwood, NY, USA).

## Biochemical analysis

The serum TG, total cholesterol (TC), and free fatty acid (FFA) were measured by an automatic biochemistry analyzer (7170; Hitachi, Chiyoda, Tokyo, Japan). Livers were homogenized at 4 °C in lysis buffer containing 50 mmol/L Tris (pH 8.0), 150 mmol/L NaCl, 1% Triton X-100, and 0.5% sodium deoxycholate. Lipids from the total liver homogenate were extracted using the chloroform/methanol method (2:1), evaporated, and dissolved in 2-propanol. Amounts of TC and TG were measured by an automatic biochemistry analyzer (7170; Hitachi 7170, Chiyoda, Tokyo, Japan).

For glucose tolerance test, mice were fasted overnight and injected with two mg/g glucose/body weight. For insulin tolerance test, mice were fasted for 4 h and injected with one U/Kg insulin/body weight. The blood glucose levels were monitored at 30 min intervals for 2 h with an ACCU-CHEK Active Blood Glucose System (Roche, Basel, Switzerland) using tail tip blood samples. Serum insulin was determined by mouse insulin ELISA kit which was purchased from R&D, Minneapolis, MN, USA.

## Cell culture

Mouse liver cancer Hepa1–6 cell lines (ATCC CRL-1830) were cultured in Dulbecco's modified eagle's medium (DMEM) (Hyclone Laboratories, Logan, UT, USA). The medium contained 10% fetal bovine serum and 100 unit/mL penicillin and 100 g/mL streptomycin in a humidified atmosphere that contained 5% $CO_2$. Cells were further administrated in DMEM containing one μM cortisol or one μM as1842856 for 48 h.

## qRT-PCR

Total RNA was isolated with Trizol reagent (Invitrogen, Carlsbad, CA, USA). cDNA was synthesized using PrimeScript[TM] RT Master Mix (Takara, Tokyo, Japan). Quantitative real-time PCR was carried out using the One Step TB Green[TM] PrimeScript[TM] RT-PCR Kit (Takara, Tokyo, Japan) according to manufacturer's instructions (Table 1). The $2^{-\Delta\Delta Ct}$ method was used to calculate the relative expression level by normalizing to β-Actin levels (Zhao et al., 2019).
**Table 1  Sequences of primers used for qRT-PCR.**

| Gene name | Sequence (5′–3′) |
| --- | --- |
| FoxO1-F1 | AGTGGATGGTGAAGAGCGTG |
| FoxO1-R1 | GAAGGGACAGATTGTGGCGA |
| Fasn-F1 | CAAGTGTCCACCAACAAGCG |
| Fasn-R1 | GGAGCGCAGGATAGACTCAC |
| Pepck-F1 | TGCGGATCATGACTCGGATG |
| Pepck-R1 | AGGCCCAGTTGTTGACCAAA |
| Scd1-F1 | GGCTAGCTATCTCTGCGCTC |
| Scd1-R1 | GAACTGCGCTTGGAAACCTG |
| HMGCoAR-F1 | ACGATCCTTCCTTATTGGCGG |
| HMGCoAR-R1 | CTCCGGATCTCAATGGAGGC |
| Fatp1-F1 | GGCAAGCTCCAGCACAGGAT |
| Fatp1-R1 | ACCCACGTACACACAGAACG |
| G6Pase-F1 | CGACTCGCTATCTCCAAGTGA |
| G6Pase-R1 | GTTGAACCAGTCTCCGACCA |
| SREBP1c-F1 | GATGTGCGAACTGGACACAG |
| SREBP1c-R1 | CATAGGGGGCGTCAAACAG |
| Cyp7a1-F1 | GGGATTGCTGTGGTAGTGAGC |
| Cyp7a1-R1 | GGTATGGAATCAACCCGTTGTC |
| Abcg1-F1 | CTTTCCTACTCTGTACCCGAGG |
| Abcg1-R1 | CGGGGCATTCCATTGATAAGG |
| Fabp1-F1 | ATGAACTTCTCCGGCAAGTACC |
| Fabp1-R1 | CTGACACCCCCTTGATGTCC |
| Cd36-F1 | ATGGGCTGTGATCGGAACTG |
| Cd36-R1 | GTCTTCCCAATAAGCATGTCTCC |
| Acc1-F1 | ATGGGCGGAATGGTCTCTTTC |
| Acc1-R1 | TGGGGACCTTGTCTTCATCAT |
| PPARα-F1 | AGAGCCCCATCTGTCCTCTC |
| PPARα-R1 | ACTGGTAGTCTGCAAAACCAAA |
| PPARγ-F1 | TCGCTGATGCACTGCCTATG |
| PPARγ-R1 | GAGAGGTCCACAGAGCTGATT |
| Pdk4-F1 | AGGGAGGTCGAGCTGTTCTC |
| Pdk4-R1 | GGAGTGTTCACTAAGCGGTCA |
| Cpt1a-F1 | CTCCGCCTGAGCCATGAAG |
| Cpt1a-R1 | CACCAGTGATGATGCCATTCT |
| Acox1-F1 | TAACTTCCTCACTCGAAGCCA |
| Acox1-R1 | AGTTCCATGACCCATCTCTGTC |
| Lcad-F1 | TCTTTTCCTCGGAGCATGACA |
| Lcad-R1 | GACCTCTCTACTCACTTCTCCAG |
| Mcad-F1 | AGGGTTTAGTTTTGAGTTGACGG |
| Mcad-R1 | CCCCGCTTTTGTCATATTCCG |
| Ucp2-F1 | TGCACTCCTGTGTTCTCCTG |
| Ucp2-R1 | GGGACCTTCAATCGGCAAGA |
| β-actin-F1 | TTCTTGGGTATGGAATCCTGT |
| β-actin-R1 | AGCACTGTGTTGGCATAGAG |

## Western blot analysis

Protein samples were prepared by lysing cells or tissues in modified RIPA buffer (1× PBS, 1% Nonidet P-40, 0.1% sodium dodecyl sulfate, and protease inhibitor cocktail (Sigma-Aldrich, St. Louis, MO, USA)). Lysates (50–100 µg) were separated on a 10% SDS-PAGE and transferred to a nitrocellulose membrane. The membrane was probed with the specific primary antibody and secondary antibody and then quantified with Odyssey Infrared Imaging System (LI-COR, Inc., Lincoln, NE, USA) and Image J Software (National Institutes of Health, Bethesda, Maryland, USA).

## Statistical analysis

The data were presented as mean ± standard error and differences considered statistically significant only when $p$-values < 0.05. Data was mainly analyzed using a two-way ANOVA followed by uncorrected Fisher's LSD post hoc test with GraphPad Prism 6 (GraphPad Software, Inc., La Jolla, CA, USA). Two-way ANOVA for repeated measures was used to analyze the weight curves. Student's $t$-test was applied to detect significant difference between two groups.

# RESULTS

## As1842856 inhibited FoxO1 activation in the liver of chronic stress mice

The levels of FoxO1 protein and mRNA were significantly increased in the liver of mice after 6 weeks of stress, while the FoxO1 inhibitor as1842856 did not affect FoxO1 expression (Figs. 1A, 1B and 1E). The ratio of $p$-FoxO1 protein to FoxO1 was significantly decreased in the stress group (Figs. 1A and 1D) but significantly increased in the stress+as1842856 group. Phosphorylated FoxO1 protein is the inactive form of FoxO1 (*Pan et al., 2017*). The changes in the mRNA levels of PEPCK and G6Pase, the classical FoxO1 regulated target genes, were consistent with the changes in the activity of FoxO1 (Figs. 1F and 1G).

## Effects of as1842856 on physiological index in mice after 6 weeks chronic stress

To investigate the effects of as1842856 on indexes of glucose metabolism and food intake, we compared these physiological indexes. During the stress period, the body weight and food intake of the control group increased slowly and steadily, while the stress group did not increase significantly. At the 6th week of stress, the body weight of the stress group was significantly lower than that of the control group (Fig. 2A). The food intake at the 5th week of stress also decreased significantly (Fig. 2B). There was no significant difference in body weight and food intake between the control group and the as1842856 group. We also tested fasting blood glucose, postprandial blood glucose, insulin, glucose tolerance, and insulin tolerance in four groups of mice after 6 weeks of stress. The results showed no significant difference between the four groups (Figs. 2C–2G).

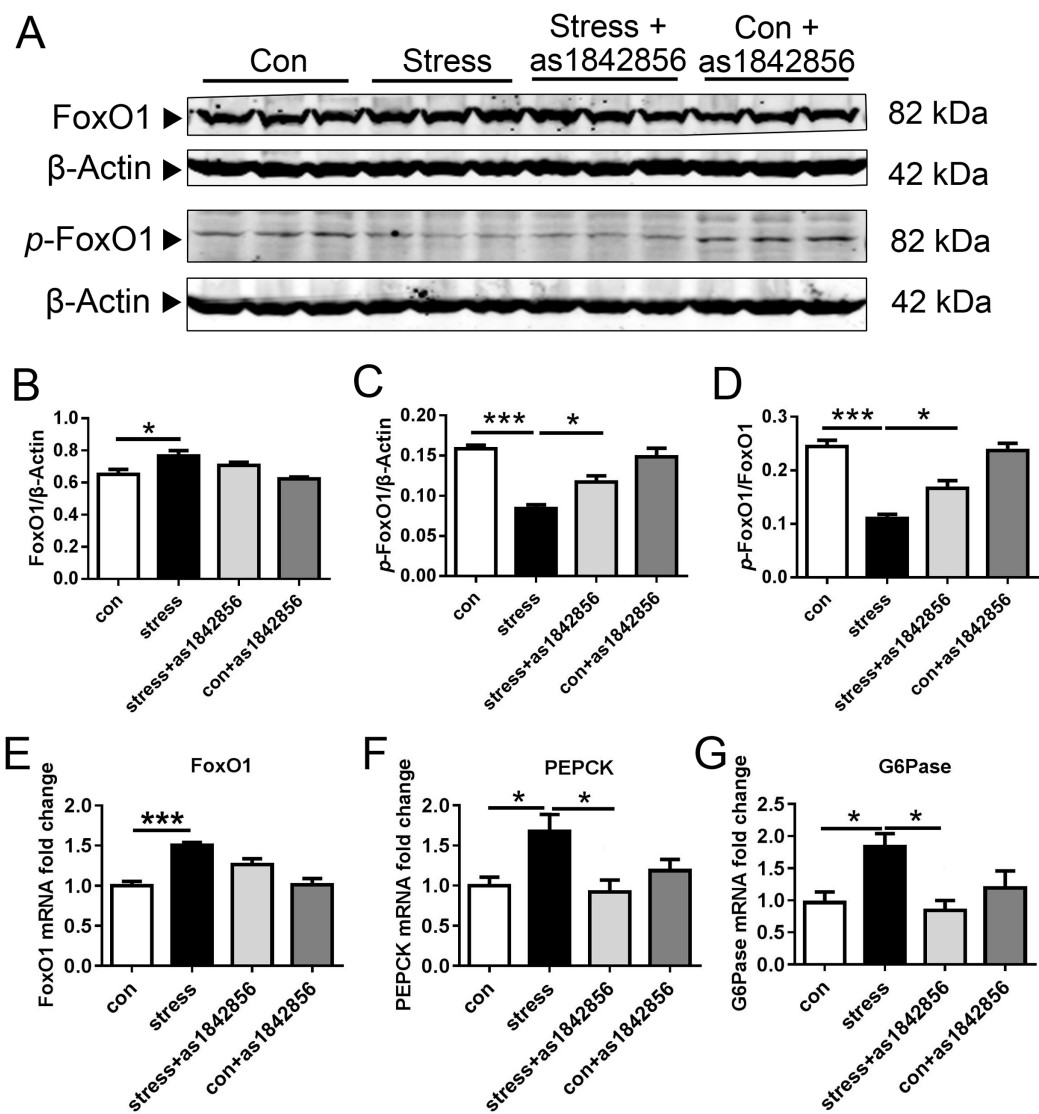

**Figure 1 FoxO1 specific inhibitor as1842856 enhances FoxO1 phosphorylation in mice liver after 6 weeks.** Chronic stress increased the protein (A and B) and mRNA lever (E) of FoxO1 expression and its downstream genes, G6Pase (F) and PEPCK (G), and decreased the lever of $p$-FoxO1 protein (C) and $p$-FoxO1/FoxO1 (D), while as1842856 could reverse this effect except the protein and mRNA lever of FoxO1 expression. $^{*}p < 0.05$, $^{***}p < 0.001$. Date presented as mean ± SEM, $n = 6$.

## As1842856 reduced adipogenesis in the liver of chronic stress mice

After 6 weeks of stress, H&E staining of livers in mice showed hepatocyte fatty degeneration: enlarged hepatocytes, loose and reticular cytoplasm, individual cytoplasm was transparent, and shaped like a balloon (Fig. 3A). Oil red O staining showed brownish red lipid droplet formation (Fig. 3A). FoxO1 inhibitor as1842856 significantly reduced the lipid deposition induced by chronic stress (Fig. 3A). Furthermore, the morphology of hepatocytes in the livers of mice fed with as1842856 still displayed normal architecture around the central veins (Fig. 3A). We further quantified the TG and cholesterol levels in

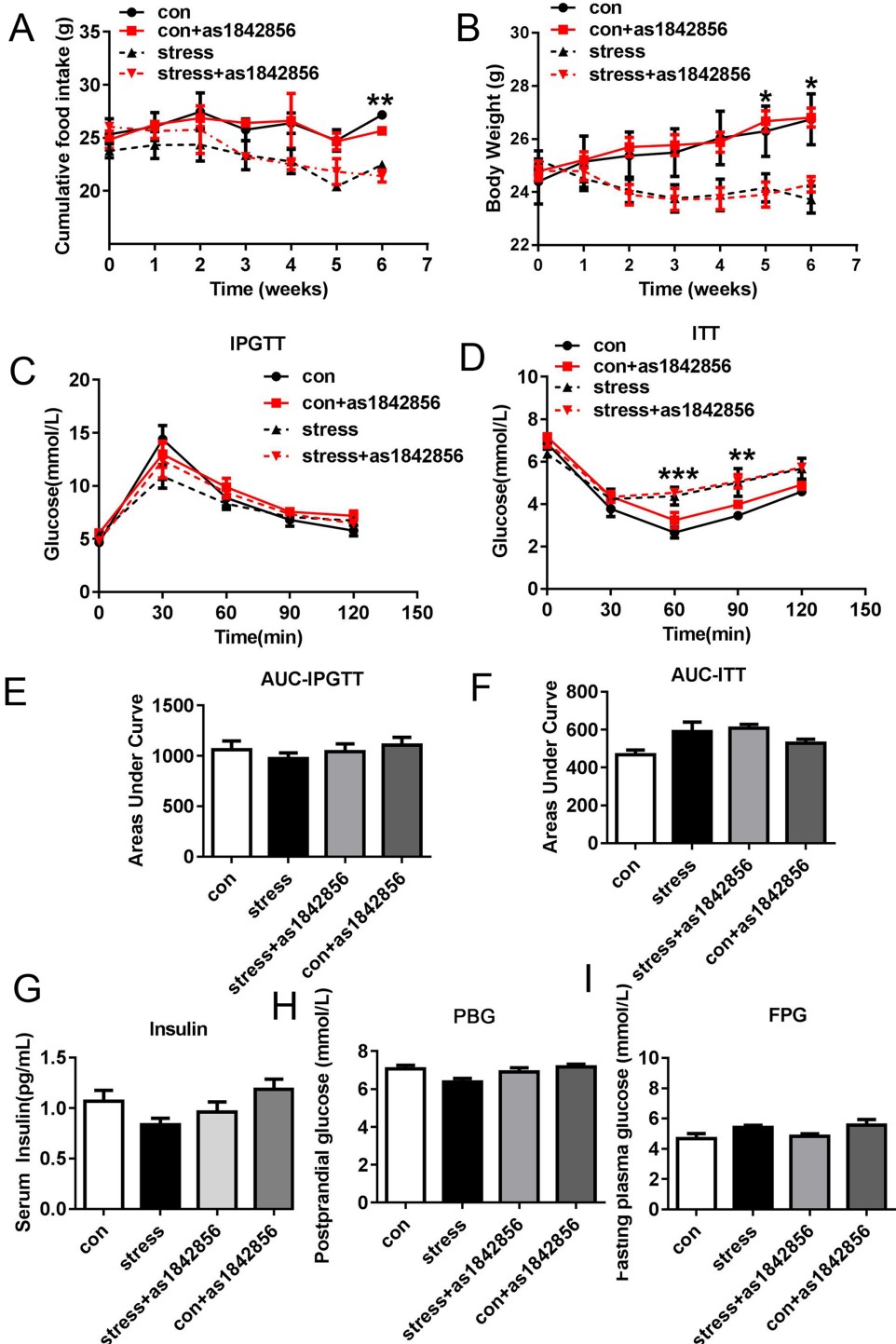

**Figure 2  As1842856 does not affect weight, food intake, and glucose metabolic parameters in mice after 6 weeks.** (A) Changes in food intake, (B) changes in body weight, (C and E) profiles of blood glucose concentration as function of time upon intraperitoneal injection of glucose, (D and F) profiles of glucose concentration (percentage of initial value) as a function of time upon intraperitoneal injection of insulin, and (G–I) blood insulin, postprandial blood glucose (PBG), and fasting plasma glucose (FPG). $^*p < 0.05$, $^{**}p < 0.01$, $^{***}p < 0.001$. Date presented as mean ± SEM, $n = 6$

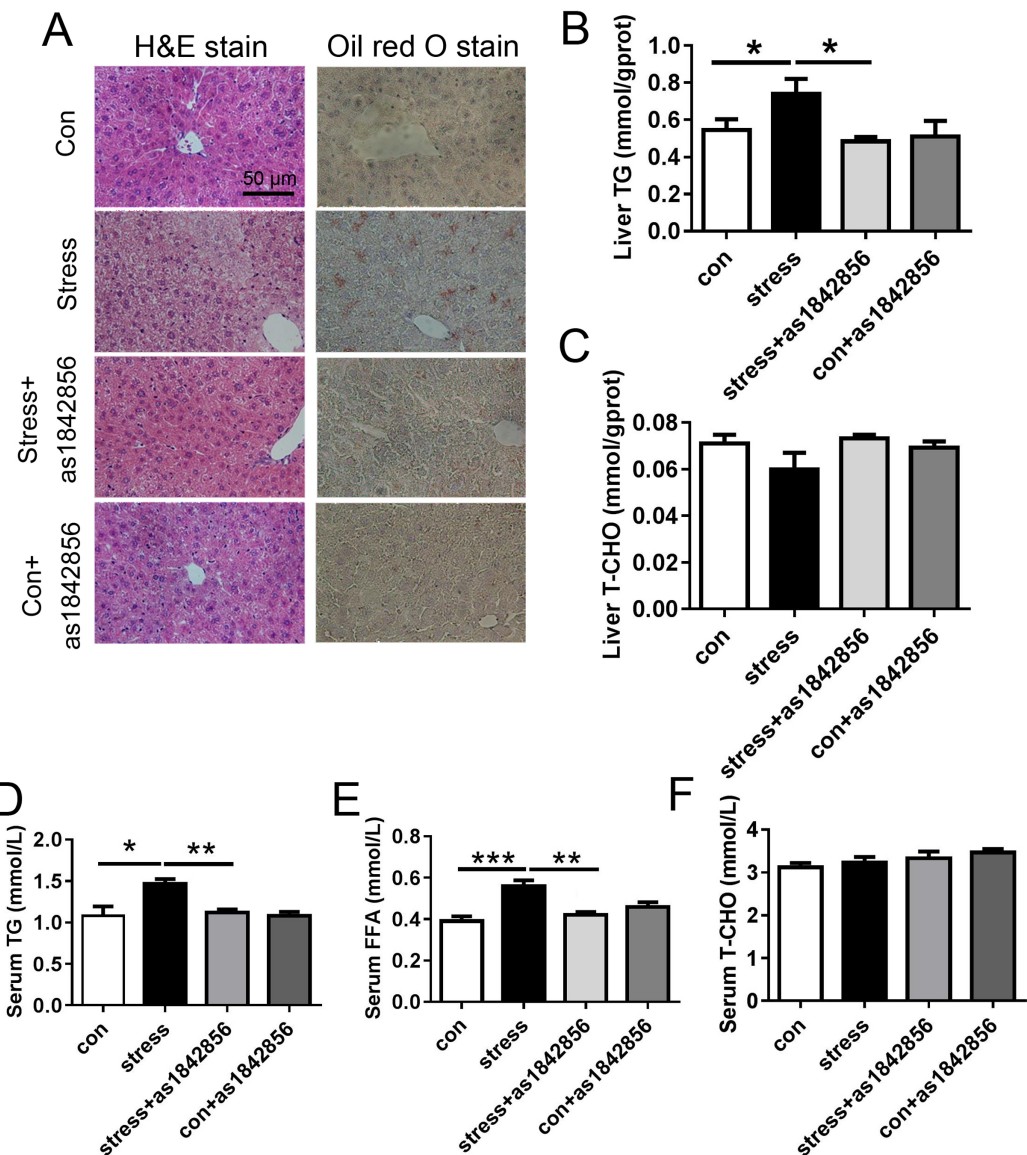

**Figure 3 As1842856 protect mice against stress-induced liver lipid deposition.** (A) Representative slides showed hematoxylin and eosin (HE)-stain and Oil red O-stain liver sections from four groups, (B and C) liver TG and T-CHO concentrations were detected by biochemical test in liver tissue homogenate, (D, E, and F) TG, FFA, and T-CHO concentrations in serum. $^*p < 0.05$, $^{**}p < 0.01$, $^{***}p < 0.001$. Date presented as mean ± SEM, $n = 6$.     

the livers of mice. The results showed that FoxO1 inhibitors significantly reduced the liver TG content induced by chronic stress, while cholesterol content had no significant change (Figs. 3B and 3C).

The levels of serum TG (Fig. 3D) and FFA (Fig. 3E, $p < 0.001$) were significantly increased in the chronic stress group. Both TG and FFA were significantly reduced in the stress+as1842856 group compared with the stress group (Figs. 3D and 3E). There was no statistical difference in serum cholesterol after chronic stress and as1842856 treatment (Fig. 3F).

### Effects of as1842856 on the mRNA expression of lipoprotein metabolism-related genes in mice after 6 weeks of chronic stress

The mRNA expression of fatty acid synthase (FASN) was significantly increased in the chronic stress group compared with the control group (Fig. 4A). However, the mRNA expression of FASN in the stress+as1842856 group was significantly decreased compared with the stress group (Fig. 4A). There were no significant differences between the four groups in ACC1 or SCD1 (Figs. 4B and 4C). The mRNA expression of fatty acid transporter protein (FATP) and fatty acid binding protein (FABP) in the chronic stress group was significantly increased compared with the control group (Figs. 4D and 4E). As1842856 significantly decreased the mRNA expression of FATP and FABP, which was elevated by the chronic stress (Figs. 4D and 4E). There were no significant differences in the other synthetically-related gene, CD36 (Fig. 4F). We also found no statistically significant differences in the mRNA expression of genes related to fatty acid oxidation, such as PPARα, Acox1, Lcad, Mcad, Pdk4, Cpt1a, and Ucp2 (Figs. 4G–4N).

### Effects of as1842856 on the mRNA expression of cholesterol metabolism-related genes in mice after 6 weeks of chronic stress

We found that the mRNA expression of HMG-CoAR and CYP7A1 was significantly increased in the stress group. The mRNA expression of HMG-CoAR and CYP7A1 in the stress+as1842856 group was significantly lower than that in the stress group (Figs. 5A and 5B). However, there was no statistical difference in the synthetic-related gene cholesterol-regulating element binding transcription factor 1C (SREBP-1c) and the cholesterol transport-related gene ABCG1 (Figs. 5C and 5D).

### As1842856 inhibited FoxO1 activation in the Hepa1–6 cells after corticosterone treatment

We also demonstrated the effect of as1842856 on FoxO1 activation and adipogenesis in the Hepa1–6 cells. The levels of FoxO1 protein and mRNA were significantly increased in the corticosterone group, compared with the control group (Figs. 6A–6C). The $p$-FoxO1 protein and the $p$-FoxO1/FoxO1 ratio were also significantly decreased after corticosterone treatment (Figs. 6A–6D). The FoxO1 inhibitor as1842856 significantly increased the $p$-FoxO1 protein and $p$-FoxO1/FoxO1 value (Figs. 6A–6D). The change of the classical target gene PEPCK regulated by FoxO1 was consistent with the change of FoxO1 activity (Figs. 6E and 6F).

### As1842856 reduced adipogenesis in the Hepa1–6 cells after corticosterone treatment

After treatment with one μM of corticosterone for 48 h, oil red O staining indicated that most Hepa1–6 cells showed obvious brown-red lipid droplets. The corticosterone +as1842856 group showed no obvious lipid droplets (Fig. 7A).

Fatty acid synthase mRNA expression was significantly increased in the corticosterone group compared with the control group (Fig. 7B). The corticosterone+as1842856 group

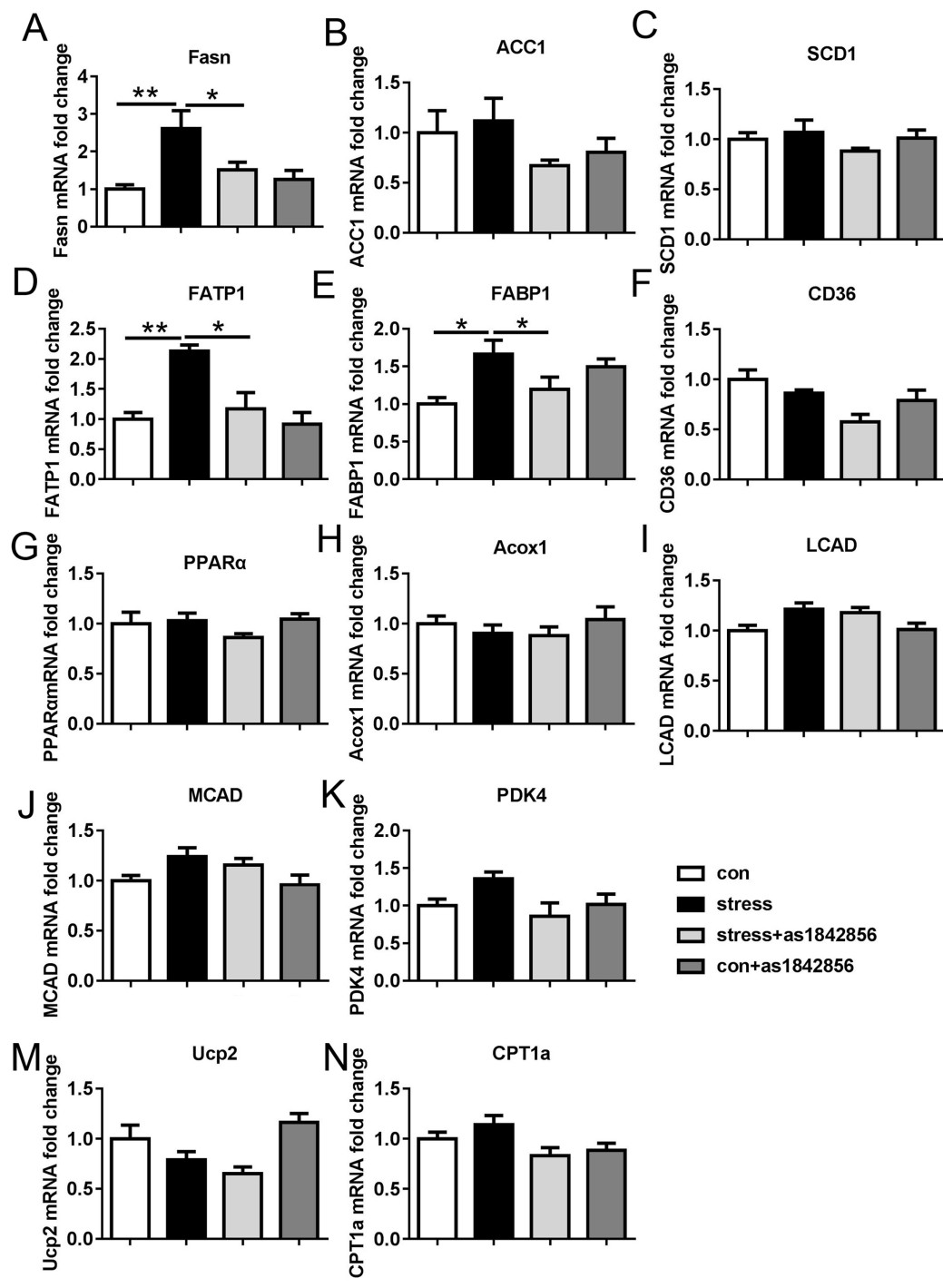

**Figure 4 Effects of stress and as1842856 on liver lipid metabolism genes in mice after 6 weeks.** (A–C) Relative mRNA levels of genes involved in hepatic TG synthesis, (D–F) relative mRNA levels of genes involved in fatty acid transport, (G–N) relative mRNA levels of genes involved in fatty acid oxidation. $^*p < 0.05$, $^{**}p < 0.01$. Date presented as mean $\pm$ SEM, $n = 6$.

showed a significant decrease in FASN mRNA expression compared with the corticosterone group (Fig. 7B). Other synthetically related genes, ACC1 and SCD1 showed no significant difference between the four groups (Figs. 7C and 7D).

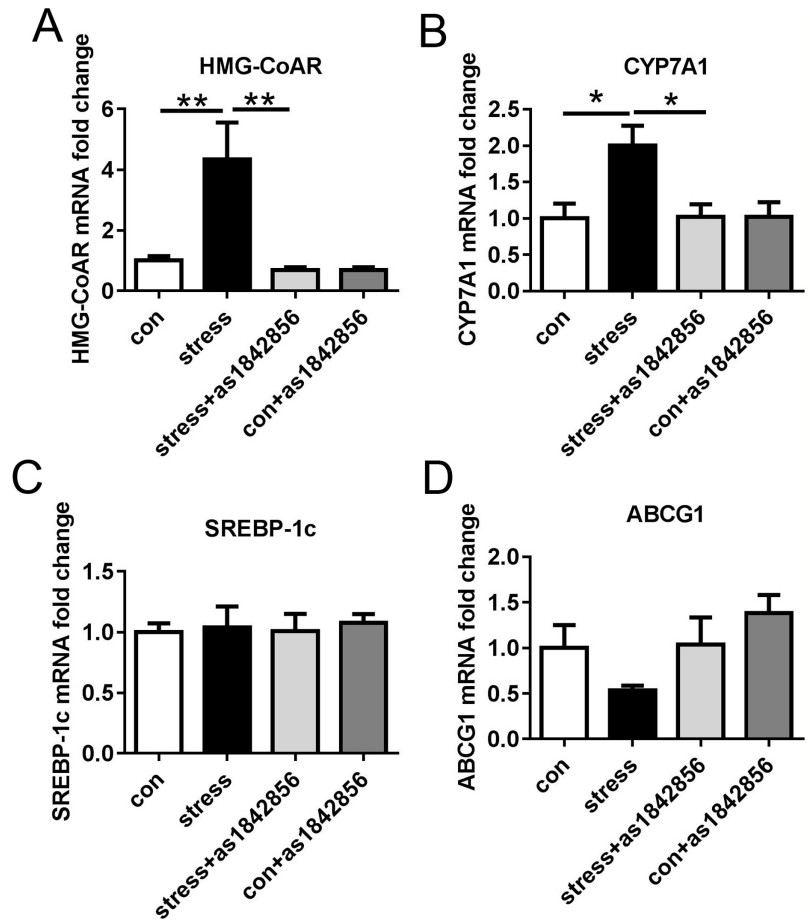

**Figure 5 Effects of stress and as1842856 on cholesterol metabolism-related genes in mice after 6 weeks.** (A–D) Liver was harvested and cholesterol metabolism-related genes were determined by real-time PCR. $^*p < 0.05$, $^{**}p < 0.01$. Date presented as mean ± SEM, $n = 6$.

## DISCUSSION

This study demonstrated that chronic stress induced lipid deposition in the livers of mice, which were mainly TG. Liver TG synthesis genes and FFA uptake genes were significantly upregulated in mRNA levels after chronic stress. These changes can be blocked by as1842856, the inhibitor of FoxO1. Moreover, at the cellular level, the changes in lipid metabolism of Hepa1–6 cells induced by corticosterone are basically the same as those in animal chronic stress experiments. The inhibitor of FoxO1 as1842856 also attenuated the lipid deposition of Hepa1–6 cells induced by corticosterone. Our study confirms the important regulatory role of FoxO1 in stress-induced lipid metabolism disorders.

Chronic stress refers to continuous physical or psychological forms of tension and pressure. *Macedo et al. (2012)* found that the chronic stress model mice, created by a narrow living environment and high-calorie diets, gained less weight than normal-fed mice. However, there was no difference between the two groups on the ratios of body weight to body length. It is suggested that mice fed a high-calorie diet may have abdominal obesity. *Czech et al. (2013)* found that chronic stress model mice caused by crowded

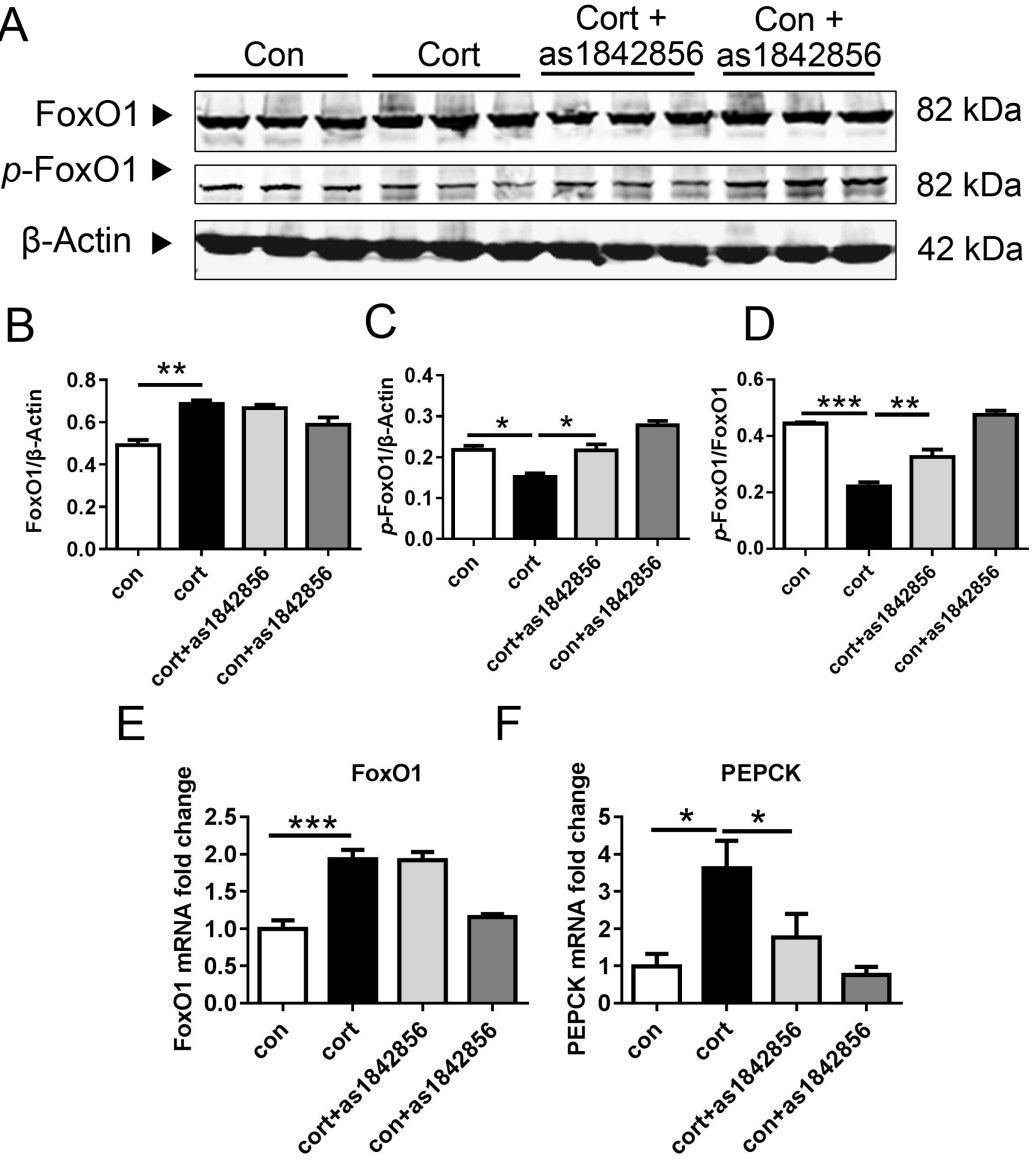

**Figure 6 As1842856 protect Hepa1–6 cells against corticosterone-induced liver lipid deposition.** Effects of corticosterone and as1842856 on FoxO1 in Hepa1–6 cells after 48 h. Corticosterone (one µM) increased the protein (A and B) and mRNA lever (E) of FoxO1 expression and its downstream gene, PEPCK (F), and decreased the lever of $p$-FoxO1 protein (C) and $p$-FoxO1/FoxO1 (D), while as1842856 (one µM) could reverse this effect except the protein and mRNA lever of FoxO1 expression. $^{*}p < 0.05$, $^{**}p < 0.01$, $^{***}p < 0.0001$. Date presented as mean ± SEM, $n = 3$.

communities have significant oxidative stress and inflammatory response in the liver under normal feeding conditions. Our previous study (*Liu et al., 2014*) found that liver index (total liver/body weight × 100) and liver TG levels increased significantly in chronic stress model (sustained foot stimulation and tethered for 12 weeks) mice. Hepatic cell adipose degeneration was found by liver section staining. Although the stress patterns and models described above are different, they all suggest that stress is associated with hepatic steatosis and inflammatory response.

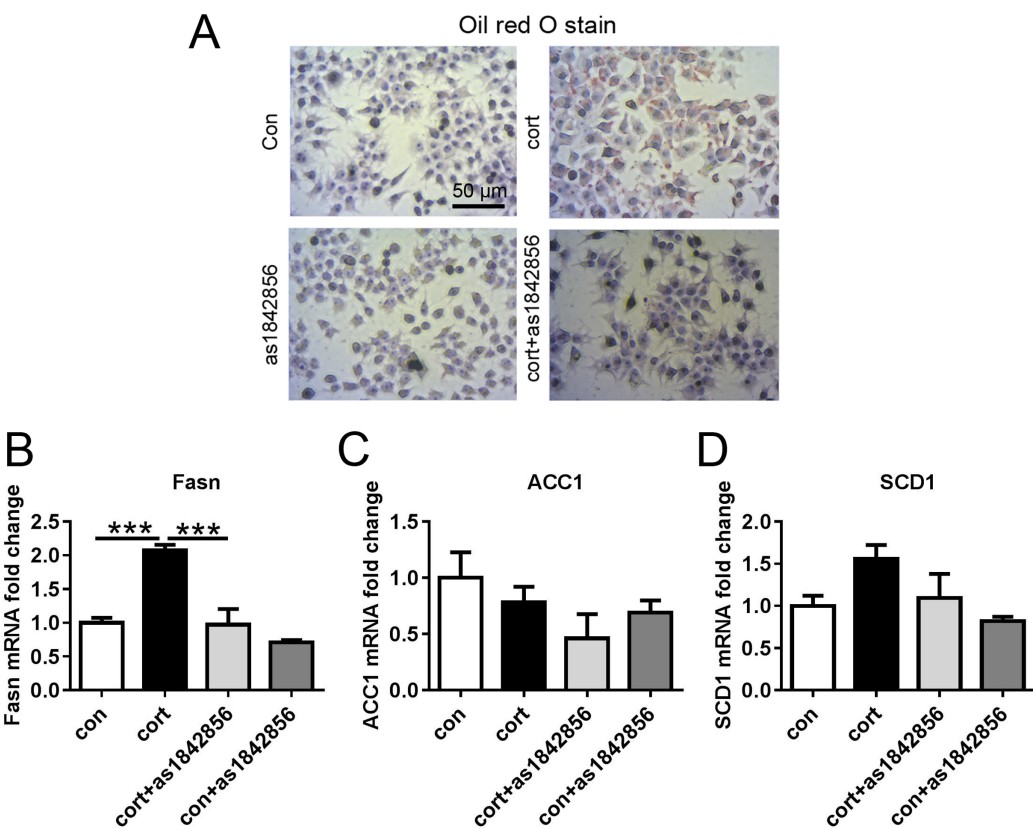

**Figure 7 Effects of corticosterone and as1842856 on lipid deposition and TG synthesis genes in Hepa1–6 cells after 48 h.** (A) Representative images showed above from four groups. After treatment with one μm Corticosterone or/and one μm as1842856 for 48 h, the cells were stained with Oil red O. (B) Corticosterone increased the genes' expression of Fasn, while as1842856 could reverse this effect. (C and D) There was no significant difference among the four groups in the genes' expression of ACC1 and SCD1. ***p < 0.001. Date presented as mean ± SEM, n = 3.

According to our previous results and related literature reports (*Kim et al., 2017*; *Shin et al., 2015*; *Yamamoto et al., 2002*), we used a combination of plantar electrical stimulation and restraint stress to construct a stress model. After 6 weeks of continuous stress, model mice liver TG levels were significantly elevated. Liver fatty degeneration was also demonstrated by the liver section staining. It indicated that chronic stress causes lipid deposition in the livers of mice, and the model was successfully constructed. It is consistent with the results of our previous study (*Liu et al., 2014*).

The possible mechanism of stress-induced hepatic lipid deposition is considered to be related to long-term activation of the hypothalamic-pituitary-adrenal cortex (HPA) axis (*Van Bodegom, Homberg & Henckens, 2017*). The activated HPA axis releases glucocorticoids. On the one hand, glucocorticoids enhance the decomposition of lipolysis hormones, promote the decomposition of adipose tissue, and release a large amount of FFA into the blood. On the other hand, glucocorticoids promote lipid formation and deposition of hepatocytes. At the same time, the hippocampus has an inhibitory effect on HPA axis activation. Under long-term chronic stress, a large number of hippocampal neurons undergo dysfunction or death by the chronically elevated glucocorticoids, losing

their inhibitory effect on the HPA axis, and leading to the massive release of glucocorticoids (*Kim, Moon & Park, 2013*; *Sakamoto et al., 2015*; *Zhu et al., 2014*). In our study, the FFA content in the blood increased significantly after chronic stress. Corticosterone or cortisol is the main hormone of the pituitary adrenocortical axis secreted by the adrenal cortex in response to environmental challenges (*McEwen, 2007*; *Quijije, 2015*; *Kant et al., 1987*). At the same time, we also investigated the effect of corticosterone in hepatocyte lipid deposition. Lipid droplets were found in Hepa1–6 cells, which were treated with corticosterone for 48 h. This suggests that chronic stress may cause liver lipid deposition by elevating the corticosterone level.

Our previous experiments found that FoxO1 may be involved in stress-induced lipid metabolism disorders. FoxO1 is a member of the FOXO transcription factor. FOXO is widely expressed in a variety of tissues, including the liver, and is a key effector molecule in cell homeostasis, metabolism, and stress response (*Ma et al., 2018*). It acts as a pre-transcriptional regulator that binds to chromatin (*Riedel et al., 2013*; *Zaret & Carroll, 2011*) and initiates or inhibits transcription (*Furuyama et al., 2000*; *Ramaswamy et al., 2002*; *Webb et al., 2013*). FoxO1 is a typical forkhead protein transcription factor, a key signaling molecule downstream of the insulin/insulin-like growth factor-1 signal and has been shown to be involved in the regulation of glucose metabolism (*O'Neill et al., 2016*). FoxO1 binds to adjacent sites in the insulin response elements within the insulin-like growth factor binding protein 1 and glucose-6-phosphatase (G6Pase) promoters to initiate gene activation (*Nakae et al., 2001a*; *Matsumoto et al., 2007*; *Yeagley et al., 2001*). Active nuclear FOXO1 also binds the transcriptional coactivator peroxisome proliferative activated receptor-γ coactivator 1-α (PGC1α) to coordinate a gluconeogenic transcriptional program involving increased expression of G6Pase and cytosolic phospho*enol*pyruvate carboxykinase (*Pck1*) (*Puigserver et al., 2003*; *Nakae et al., 2001b*). In addition, FOXOs mediate the effects of insulin on adipocyte differentiation, neuropeptide transcription and processing, and β-cell health (*Nakae et al., 2003*; *Ren et al., 2012*; *Plum et al., 2009*). In recent years, an increasing number studies have shown that FoxO1 is also involved in the regulation of lipid metabolism in the liver. These studies have found that activation of FoxO1 was observed in lipid deposition in the liver (*Kim et al., 2016*); activated FoxO1 induced an increase in liver output of very low-density lipoprotein (rich in TG particles) and hypertriglyceridemia (*Kim et al., 2011*). This study also indicated that FoxO1 does participate in liver lipid deposition caused by chronic stress. On the one hand, we observed changes in FoxO1 protein, mRNA, and phosphorylation levels in stressed mice; on the other hand, liver lipid deposition due to chronic stress is alleviated after the treatment of the FoxO1 activity inhibitor.

To further clarify how FoxO1 regulates liver lipid deposition, this experiment examined the expression levels of genes involved in liver and lipid metabolism. Previous experiments have confirmed that the main component of lipid deposition caused by chronic stress is TG, and the amount of TG in the liver is inseparable from the uptake and synthesis of FFA (*Liu et al., 2014*). FASN is a key enzyme for the de novo synthesis of endogenous fatty acids and is closely related to lipid metabolism (*Jones & Infante, 2015*). FASN mRNA and protein levels were significantly upregulated in fatty liver (*Huang, Gusdon & Qu,*

*2013*). Upregulation of FASN in rat liver is involved in the formation of fatty liver. The FATP family and the FABP family are important protein families that take up fatty acids and are primarily responsible for the transport of extracellular FFA uptake. Studies have confirmed that fatty acid uptake is significantly reduced in hepatocytes of FATP knockout mice (*Falcon et al., 2010*). Purified FATP1 exhibits long-chain and ultra-long-chain fatty acyl-CoA synthetase activity (*Hall, Smith & Bernlohr, 2003*). FABP can be activated by long-chain FA, PPARs, and toll-like receptor agonists, which are involved in sugar, lipid metabolism, and inflammatory processes (*Doege et al., 2006*; *Tan et al., 2002*; *Wolfrum et al., 2001*). This study found that FoxO1 increases fatty acid synthesis and intra-hepatic transport of FFA by increasing the expression of FASN, FATP, and FABP and increasing liver TG deposition. ACC1 and SCD1 are two other key enzymes involved in lipogenesis (29925265). ACC1 and SCD1 showed no significant difference between the four groups. This suggested that the expression of FASN, FTAP, and FABP, but not ACC1 or SCD1, are controlled by the transcription factor FoxO1. We also found that FoxO1 did not affect the feeding of mice and fatty acid oxidation in the liver. FoxO1 is a key molecule downstream of insulin, which regulates glucose metabolism. However, chronic stress mice did not show significant glucose metabolism abnormalities in this experiment. We speculate that mice may still be in a compensatory period due to the relatively short stress time. This conclusion requires further experiments to prove by a longer-term chronic stress test.

## CONCLUSIONS

In summary, this study demonstrated that chronic stress induced FoxO1 activation and lipid deposition in the livers of mice. Inhibition of FoxO1 attenuated the TG synthesis and fatty acid oxidation induced by the chronic stress. Corticosterone may act as a mediator between FoxO1 and chronic stress. The present study indicated that inhibition of FoxO1 may have therapeutic benefits for chronic stress relative fatty liver disease.

### Funding

This research was fully funded by the Natural Science Fund of Shanghai (17ZR1437800) (Yun-zi LIu), the National Natural Science Foundation of China (NSFC) (31871171 and 81571169) (Chun-lei Jiang), and (81703267) (Ji-kuai Chen). The funders had no role in study design, data collection and analysis, decision to publish, or preparation of the manuscript.

### Grant Disclosures

The following grant information was disclosed by the authors:
Natural Science Fund of Shanghai: 17ZR1437800.
National Natural Science Foundation of China (NSFC): 31871171, 81571169, and 81703267.

## Competing Interests

The authors declare that they have no competing interests.

## Author Contributions

- Yun-zi Liu conceived and designed the experiments, performed the experiments, prepared figures and/or tables, authored or reviewed drafts of the paper, approved the final draft.
- Wei Peng conceived and designed the experiments, performed the experiments, prepared figures and/or tables, authored or reviewed drafts of the paper, approved the final draft.
- Ji-kuai Chen performed the experiments, approved the final draft.
- Wen-jun Su analyzed the data, approved the final draft.
- Wen-jie Yan contributed reagents/materials/analysis tools, approved the final draft.
- Yun-xia Wang approved the final draft.
- Chun-lei Jiang conceived and designed the experiments, authored or reviewed drafts of the paper, approved the final draft.

## Animal Ethics

The following information was supplied relating to ethical approvals (i.e., approving body and any reference numbers):

All animal procedures used in this study were approved by the Institutional Animal Care and Use Committee of the Second Military Medical University (No. 20161027, Shanghai, China) and the Shanghai Science and Technology Committee (SYXK-HU-2012-0003).

## Data Availability

The raw measurements are available in the Supplemental Files.

## Supplemental Information

Supplemental information for this article can be found online at http://dx.doi.org/10.7717/peerj.7668#supplemental-information.

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
