# Peer review of "FoxO1 is a critical regulator of hepatocyte lipid deposition in chronic stress mice"

_PeerJ, doi:10.7717/peerj.7668_

## Round 0.1 · original submission · Major Revisions

Please revise the manuscript according to the comments

[]

Reviewer 1 ·

Basic reporting

The use of English is clear, though there are plenty of grammar mistakes await the author to correct. Introduction and background are included, and structure seems in discipline. Figure 1A and 6A doesn’t seem qualified for publication, and there are errors in legend of these two figures. Raw data supplied.

Experimental design

It is an original primary research within scope of the journal. Methods are all sufficiently described, and conducted in conformity with the prevailing ethical standards in the field. The scientific hypothesis of this article is clearly proposed, and the question is quite meaningful, however, the elucidation may be incomprehensive.

Validity of the findings

Conclusions are well stated, linked to original research question,and limited to supporting results.

Additional comments

[1] The most important issue of this article is that although you claimed to bring the molecular mechanisms of stress-induced lipid metabolism disorders some clarity, however, the design of the whole experiment only explore the result of inhibiting FoxO1, which is hard to call a mechanism.

[2] The language and the style of writing certainly need improvement. The abstract should be a synopsis, which means you should summarize everything rather than present what and why you did in simple sentences. Similar language problems happen in the result section, where logic between each figure should be slightly elucidated, or readers may be easily lost track of your purpose. Furthermore, the order you present your result might need to be in line with how you put it in discussion, in this way you can bring unification into your article. Obvious grammar mistakes can be found, please correct them all before publication.

[3] The figures 1A and 6A can hardly be considered qualified. I checked the original pictures in raw data, still, the shadows and dots can be a real problem for analyzing, and the actin in 6A doesn’t seem like it’s on a straight line.

[4] NAFLD is a major risk factor for the development of type 2 diabetes and a central feature of the metabolic syndrome(Asrih & Jornayvaz, 2013). There are literatures featured in FoxO1, adipocyte differentiation, and its regulation by insulin(Puigserver et al., 2003)(Nakae et al., 2003), and certain mechanisms are presented. However, such progresses are not mentioned well enough in this article, except in line 247 and ITT. However, in line 247, that’s just an introduction of FoxO1, and ITT can merely explain a mechanism. Clearing this may help valid your results.

[5] The introduction of corticosterone is abrupt, and if you want to call it a model, please verify it or provide with previous studies. And your corticosterone treatment was only applied to a cell line, but if you want to discuss the HPA axis, in vivo treatment may be better.

[6] In your methods, you mention that your data was mainly analyzed using one-way ANOVA. However, you are using corticosterone and as1842856 treatments, and they are two different effectors, thus the statistical approach should be two-way ANOVA. The significance may alter with the change of analyzing methods, please change it in modification.

[7] Titles of figures are overly narrative, please summarize your theme of each figure.

Reviewer 2 ·

Basic reporting

In this paper, Liu and colleagues report that FoxO1 is a critical regulator of hepatocyte lipid deposition in chronic stress mice, extending their previous findings. The results are interesting and the article is valuable for publication. However, some small concerns need to be addressed.
1) Use consistent abbreviations in the manuscript, e.g. TG for triglyceride, FFA for free fatty acid (FFA), etc.
2) It’s better to include all the used reagents into Section “Reagents and Antibodies” in Materials & Methods.
3) Line 78: The as1842856 was administered by gavage, whether this kind of treatment causes additional stress to the animals?
4) Line 124: Check “Bonferroni’s multi-ple comparisons”.
5) Line 188: “After treated with 1 μm corticosterone for 48 h”. Does it mean 1 μM?
6) Line 260-263: Include the reference.
7) The authors mentioned that Other synthetically related genes, ACC1 and SCD1 showed no significant difference between the four groups, it’s better to give one or two sentences to discuss this observation.

Experimental design

No comment.

Validity of the findings

No comment.

Additional comments

None.

Reviewer 3 ·

Basic reporting

English needs polishing furhter.

Experimental design

This paper raised a good question, the experimental design is OK, but the data set are kind of worrysome.

Validity of the findings

See the specific comments to author below.

Additional comments

Liu et al. in their manuscript entitled “Foxo1 is a critical regulatory of hepatocyte lipid deposition in chronic stress mice” tried to come up with a mechanistic explanation for how Foxo1 regulates chronic stress-induced liver lipid deposition, a basis for the nonalchoholic fatty liver diseases (NAFLD). It is potentially interesting. The data set need to be improved and the writing need more editing and enhancement though.

Specific comments:

1. Does as1842856 has any effect on the morphology and function of the liver? In figure2A, the picture of con and con+as1842856 looks different.
2. Despite that as182856 does have effect on Foxo1 phosphorylation and target gene expression, it does not rescue or change the effect of stress condition on body weight loss at all (Figure B). This is an interesting puzzle and should be explained or speculated.
3. In results, the description on figure3A and 3B seems to be mistaken. Also in figure3B, the stress+as1842856 group is dramatically different from control, they are not consistent with what is described in the paper.
4. Why as1842856 has no effect on basal Foxo1 activity under steady condition?
5. The English language should be improved throughout the manusript.

---

## Round 0.2 · accepted · Accept

Thanks for improving the manuscript.

Reviewer 1 ·

Basic reporting

no comment

Experimental design

no comment

Validity of the findings

no comment